# A *sart1* Zebrafish Mutant Results in Developmental Defects in the Central Nervous System

**DOI:** 10.3390/cells9112340

**Published:** 2020-10-22

**Authors:** Hannah E. Henson, Michael R. Taylor

**Affiliations:** 1Chemical Biology and Therapeutics Department, St. Jude Children’s Research Hospital, Memphis, TN 38015, USA; 2College of Graduate Health Sciences, University of Tennessee Health Science Center, Memphis, TN 38163, USA; 3School of Pharmacy, University of Wisconsin-Madison, Madison, WI 53705, USA; michael.taylor@wisc.edu

**Keywords:** *sart1*, zebrafish, RNA-Seq, central nervous system

## Abstract

The spliceosome consists of accessory proteins and small nuclear ribonucleoproteins (snRNPs) that remove introns from RNA. As splicing defects are associated with degenerative conditions, a better understanding of spliceosome formation and function is essential. We provide insight into the role of a spliceosome protein U4/U6.U5 tri-snRNP-associated protein 1, or Squamous cell carcinoma antigen recognized by T-cells (Sart1). Sart1 recruits the U4.U6/U5 tri-snRNP complex to nuclear RNA. The complex then associates with U1 and U2 snRNPs to form the spliceosome. A forward genetic screen identifying defects in choroid plexus development and whole-exome sequencing (WES) identified a point mutation in exon 12 of *sart1* in *Danio rerio* (zebrafish). This mutation caused an up-regulation of *sart1*. Using RNA-Seq analysis, we identified additional upregulated genes, including those involved in apoptosis. We also observed increased activated *caspase 3* in the brain and eye and down-regulation of vision-related genes. Although splicing occurs in numerous cells types, *sart1* expression in zebrafish was restricted to the brain. By identifying *sart1* expression in the brain and cell death within the central nervous system (CNS), we provide additional insights into the role of *sart1* in specific tissues. We also characterized *sart1*’s involvement in cell death and vision-related pathways.

## 1. Introduction

Although the splicing process was discovered more than 30 years ago [1], the mechanisms of spliceosome assembly and splicing regulation are not well understood. To date, over 100 proteins that regulate spliceosome assembly and function have been identified [2]. Understanding the role of these proteins in splicing is important because several disorders such as retinitis pigmentosa (RP) [3], amyotrophic lateral sclerosis, spinal muscular atrophy [4], and chronic lymphocytic leukemia [5] are caused by mutations in splicing factors.

We have characterized one of these genes, *sart1*, which was isolated from an N-ethyl-N-nitrosourea (ENU) mutagenesis forward genetic screen in the enhancer trap zebrafish line *Et*(*cp:EGFP*)*^sj2^* to identify mutants in choroid plexus (CP) development [6]. In this genetic screen, zebrafish larvae were observed at 4 days postfertilization (dpf) to look for phenotypes in the CP. The green fluorescent protein (GFP), was expressed in CP epithelia so mutants in CP development could be easily observed under a fluorescent microscope. Mutants identified from the screen were classified based on intensity of GFP expression in the CP, localization of epithelial cells, and overall CP size. Previously referred to as *cp27.5*, this line with a mutation in *sart1* had variable GFP expression, small epithelial aggregates, and an expanded CP. The mutants also had expanded ventricles, small eyes, and increased CNS barrier permeability. Mutants were embryonic lethal by late day 4. Using bulked segregant analysis, we were able to map the mutation to chromosome 21. In our current study, we identified the mutation in *sart1* using exome sequencing, which has been utilized as an effective and time-efficient strategy to identify mutations from lines discovered in zebrafish forward genetic screens [7].

Sart1, also known as U4/U6.U5 tri-snRNP-associated protein 1, squamous cell carcinoma antigen recognized by T-cells, HAF in mice, and Snu66 in yeast, is a complex protein involved in U4/U6.U5 tri-sRNP recruitment to the spliceosome [8]. There are two protein isoforms of Sart1, one is an 800 amino acid sequence located in the nucleus of proliferating cells and contains a leucine zipper motif suggesting it may bind to DNA [9]. Another 259 amino acid protein is synthesized by a frame shift in an internal ribosomal entry site that has been shown to be expressed in the cytosol in squamous cell carcinomas [9].

While splicing genes are thought to be ubiquitously expressed, recent evidence suggests that certain components of the spliceosome may be temporally or spatially regulated. Studies have identified spliceosome components such as U2 that have high RNA levels, specifically in the cerebellum, and mutations in U2 result in neurodegeneration [10]. In zebrafish, we observed that *sart1* is expressed specifically in the brain and upregulated in *cp27.5* mutants. This provides further evidence that splicing components may be regulated or expressed in specific tissues resulting in tissue-specific phenotypes.

In addition, we have demonstrated that this mutation results in up or downregulation of numerous gene classes. Using RNA-Seq analysis, we identified that *sart1* is upregulated in mutants. It also results in upregulation of genes essential to the apoptotic pathway such as *tumor protein 53* (*tp53*) and the proto-oncogene *mdm2*. Conversely, this mutation causes genes regulating eye development and function to be downregulated. This analysis confirmed our observations of retinal lamination loss and increased cell death in the eyes of *cp27.5* mutants. We also observed an upregulation of other spliceosome components such as *sm-like 7* (*lsm7*) and *pre-mRNA processing factor 31* (*prpf31*) which may indicate a compensation mechanism due to defects in *sart1* function. While previous studies have identified the major role of *S*art1 as recruiting the U4/U5/U6 tri-sRNP to the spliceosome, we provide evidence that it may directly affect specific classes of genes such as those involved in apoptosis along with eye formation and maintenance. Understanding *S*art1 regulation of gene expression will reveal how spliceosome components not only play a role in splicing, but also other cellular processes, such as apoptosis, and pathological conditions, such as degeneration and cancer.

## 2. Materials and Methods

### 2.1. Isolating cp27.5 Mutants

*cp27.5* mutants were previously identified from an ENU mutagenesis screen identifying mutants in CP development. CP morphology and function of *cp27.5* mutants were characterized by Henson et al. [6].

### 2.2. Fish Lines and Maintenance

Zebrafish were maintained in accordance with established protocols and all experiments were approved by the St. Jude Children’s Research Hospital Institutional Animal Care and Use Committee. Zebrafish were maintained at 28.5 °C on a 14 h light/dark cycle. Embryos used for imaging were collected in egg water (0.03% Instant Ocean) and treated at 24 h postfertilization (hpf) with 0.003% 1-phenyl-2-thiourea (PTU) (Sigma) to prevent pigment formation. Fish strains used for this study include *Et*(*cp:EGFP*)*^sj2^*, *cp27.5*, and TL.

### 2.3. WES and Analysis

*cp27.5* wild-type and mutant larvae (20 each) were collected in MeOH at 3 dpf. The 20 larvae were generated from 3 different heterozygous parents. DNA was extracted using the MagAttract High Molecular Weight (HMW) DNA kit (Qiagen Sciences LLC, Germanton, MD, USA) following the manufacturer’s instructions. Values for DNA concentration and purity were obtained by a ND-1000 NanoDrop Spectrophotometer and Qubit assay (Invitrogen, Carlsbad, CA, USA). Concentration was adjusted to the required 3 µg/120 µL for sequence capture. Sequence capture and sequencing was done by the Genome Sequencing Facility Hartwell Center for Bioinformatics and Biotechnology at St. Jude Children’s Research Hospital. Paired end 100 cycle sequencing was run on a HiSeq 2500 using TruSeq SBS v3 chemistry according to the manufacturer’s instructions (Illumina, San Diego, CA, USA).

From the raw sequencing data, the quality control, read mapping, and variant calling were performed using CLC Genomic Workbench v6.5 (CLC Bio, Aarhus, Denmark). In brief, the reads were trimmed against the sequencing adapters, and only reads with a sequencing quality greater than 20 and a read length greater than 50 bp were retained. The filtered reads were then aligned to the zebrafish reference genome sequence (Zv9 assembly, 2010), and the lists of single-nucleotide variants (SNV) and indels were generated. To identify the causative mutation responsible for the phenotype, we compared the variants of mutant with those of wild type, and a region enriched with mutant-specific homozygosities was observed at a peak of homozygosity scores defined by the percentage of mutant-specific homozygous variants within a 1-Mb window across the genome. A candidate mutation was further pinpointed by screening the region for nonsense and essential splice mutations. A diagram of whole-exome sequencing (WES) workflow and analysis is represented in Figure 1.

### 2.4. Reverse Transcriptase (RT) PCR

Larvae were collected on ice in 50 µL Trizol (Life Technologies, Carlsbad, CA, USA) and stored at −80 °C until use. Samples were homogenized and a phenol-chloroform extraction was used to isolate RNA. After the RNA precipitation using isopropanol, the SuperScript^®^ III Reverse Transcriptase kit (Life Technologies, Carlsbad, CA, USA) was used according to the manufacturer’s instructions to prepare cDNA. cDNA was diluted 1:5 in water for PCR. RT-PCR was done using the AccuPrime Taq DNA Polymerase System (Invitrogen, Carlsbad, CA, USA). Samples were analyzed on a 1% agarose gel. Primer sequences included: *sart1*F1: 5′-CGTTTTTAAGCCAAAGTCTGTGCTG-3′; *sart1*R1: 5′-CACCTCCTTCTTTCTCGTCATCCTT-3′.

### 2.5. sart1 mRNA Rescue

cDNA from 3 dpf wild-type larvae was amplified using the AccuPrime Taq DNA Polymerase System to produce a ~2.5 kb fragment of *sart1* open reading frame (ORF). The *sart1* fragment was cloned into the PCRII TOPO vector using the PCRII TOPO TA cloning kit dual promoter (Life Technologies, Carlsbad, CA, USA) and transformed into TOP 10F’ chemically competent *Escherichia coli* (Invitrogen, Carlsbad, CA, USA). Plasmid DNA was purified using the DNA Miniprep kit (Qiagen Sciences LLC, Germantown, MD, USA). The plasmid was then cut using *Eco*RV and *Spe*1 restriction enzymes (New England BioLabs, Ipswich, MA, USA) to release the *sart1* fragment. The PCS2^+^ vector was cut using *Sna*B1 and *Xba*1 and a ligation reaction was performed to insert *sart1* into the PCS2^+^ vector. After molecular cloning of PCS2^+^:*sart1*, DNA was purified using the DNA Miniprep Kit (Qiagen Sciences LLC, Germantown, MD, USA). Purified DNA was linearized using *Not*1 (New England BioLabs) and RNase A was added to the reaction.

Following the protocols as outlined in the mMessage machine kit (Ambion, Life Technologies, Austin, TX, USA), transcription-quality plasmid DNA was prepared and RNA was synthesized from the linearized DNA plasmid. For the rescue, approximately 150 pg of *sart1* mRNA was injected into single-celled embryos produced from *cp27.5* heterozygous parents. Embryos were observed daily to look for rescue of the mutant phenotype. A scoring system based on the phenotype at 4 dpf was used to determine the rescue efficiency. Larvae were scored as wild-type, mutant, or abnormal. Embryos were considered abnormal if there was an observable defect that did not resemble a wild-type or mutant phenotype. After scoring the larvae based on phenotype, samples were also collected and genotyped using PCR and Sanger DNA sequencing to determine how many mutant larvae were rescued. The percentage rescued was based on the number of mutants (based on genotype) that had wild-type phenotypes. Abnormal phenotypes for mutants were considered a partial rescue since most “abnormalities” were minor compared to phenotypes of uninjected mutants, such as reduced head size or slight heart edema.

### 2.6. Whole Mount In Situ Hybridization

The RNA probe was synthesized from the PCRII TOPO vector containing the 2.5 kb *sart1* fragment as mentioned above using the DIG RNA Labeling Kit (SP6/T7) (Roche, Basel, Switzerland). A probe for *deltaC* was also synthesized to use as a control. The probes were purified using the illustra Probe Quant G-50 micro columns (GE Healthcare, Chicago, IL, USA) and resuspended in 80 µL hybridization buffer. Hybridization buffer was prepared as described in Thisse and Thisse [11]. Probes were analyzed on a 1% agarose gel to determine the quality of the RNA. *cp27.5* wild-type and mutant larvae were collected at 3 dpf, anesthetized in 0.02% tricaine, and fixed in 4% paraformaldehyde (PFA) (Electron Microscopy Sciences, Hatfield, PA, USA). Larvae were stored at 4 °C for 16 h overnight. The whole mount in situ hybridization protocol was performed as previously described in Thisse and Thisse [11].

### 2.7. Immunohistochemistry (IHC)

Larvae were anesthetized in 0.02% tricaine and fixed in 4% PFA at 4 °C overnight and washed the next day in 1× phosphate buffered serum (PBS) (Calbiochem, San Diego, CA, USA). Samples were sunk in 30% sucrose/PBS at 4 °C overnight and embedded in Tissue-Tek Optimal Cutting Temperature (O.C.T.) Compound (Sakura-Finetek, Torrance, CA, USA), frozen on dry ice, and stored at −80 °C. Tissue sections were washed in PBS and PBST [PBS/0.03% Triton X-100 (Sigma-Aldrich, St. Louis, MO, USA)] and incubated in blocking buffer (PBST with 5% goat serum (Gibco, Dublin, Ireland) and 1% BSA (Sigma-Aldrich, St. Louis, MO, USA)). Primary antibodies were incubated at 4 °C overnight followed by secondary antibody incubation for 2 h at room temperature. Primary antibodies included rabbit antiactivated Caspase 3 (1:200; Cell Signaling Technology, Danvers, MA, USA), rabbit anti-Glut1 (1:200, Novus Biologicals, Littleton, CO, USA), mouse anti-SV2 (1:300 Developmental Studies Hybridoma Bank, Iowa City, IA, USA), mouse anti-Zpr1 (1:100; ZIRC, Eugene, OR, USA), mouse anti-HuC/HuD neuronal protein (1:500; Life Technologies, Carlsbad, CA, USA), and rabbit anti-S100β (1:1000; Dako, Carpinteria, CA, USA). Secondary antibodies included Alexa Fluor goat antirabbit 488 (1:200; Invitrogen, Carlsbad, CA, USA) and Alexa Fluor goat antimouse 555 (1:200; Invitrogen, Carlsbad, CA, USA). Sections were counterstained with 1 µg/mL DAPI (Roche, Basel, Switzerland). Images were taken on a Nikon E800 microscope and analyzed using NIS-Elements AR Version 3.2.14 software.

### 2.8. Whole Mount IHC

Embryos were incubated in egg water with 0.003% PTU to prevent pigment formation. Larvae were anesthetized in 0.02% tricaine and fixed in 4% PFA overnight. Samples were washed in 1× PBS followed by 1× PBST and treated with 20 µg/mL Proteinase K (New England Biolabs, Ipswich, MA, USA). The reaction was stopped by adding 10% lamb serum (Gibco, Dublin, Ireland) followed by additional washes in PBST. Samples were blocked with 10% lamb serum for 1–4 h and incubated in primary antibodies followed by secondary antibodies. Antibodies used included rabbit anti-GFP (1:100; Invitrogen, Carlsbad, CA, USA), rabbit anti-activated Caspase-3 (1:100; Cell Signaling Technology, Danvers, MA, USA), mouse anti-Zpr1 (1:50; ZIRC, Eugene, OR, USA), and mouse anti-Cldn5 (1:100; Invitrogen, Carlsbad, CA, USA), Alexa Fluor goat anti-rabbit 488 (1:200; Invitrogen, Carlsbad, CA, USA) and Alexa Fluor goat anti-mouse 555 (1:200; Invitrogen, Carlsbad, CA, USA). Samples were imaged on Nikon C1Si laser scanning confocal microscope. Z-stacks were compiled to create maximum intensity projection images using Nikon NIS-Elements Version 3.1 imaging software.

### 2.9. Western Blot

An equal number of wild-type and mutant larvae were each collected in homogenization buffer (2×) (Mini Complete tablet, Roche, Basel, Switzerland) containing 0.002% DNase (Ambion Life Technologies, Austin, TX, USA). After homogenization, SDS/gel loading buffer containing DTT was added to the homogenate. Samples were then placed in boiling water for 5 min. Samples and the Precision Plus Kaleidoscope Protein Standard ladder (BioRad, Hercules, CA, USA) were run on a 4–15% polyacrylamide gel (BioRad, Hercules, CA, USA). Proteins were transferred to a nitrocellulose membrane using the Trans-Blot Turbo Transfer System (BioRad, Hercules, CA, USA). Following protein transfer, the nitrocellulose membrane was blocked in 5% milk. Membranes were incubated in primary antibody at 4 °C overnight. The membrane was then washed in 1× Tris-buffered saline/0.1% Tween (TBST) at RT and incubated in secondary antibody for 1 h at RT. After incubation, the membranes were washed in 1× TBST and imaged using LI-COR Odyssey 1393 and Odyssey Infrared Imaging System Version 3.0. Primary antibodies included rabbit anti-p53 (1:500; GeneTex, Irvine, CA, USA) and mouse anti-actin (1:1000; Sigma-Aldrich, St. Louis, MO, USA). Secondary antibodies included goat anti-rabbit IRDye ^®^ 680 (1:20,000; LI-COR Biosciences, Lincoln, NE, USA) and goat anti-mouse IRDye ^®^ 800 CW (1:15,000; LI-COR Biosciences, Lincoln, NE, USA).

### 2.10. Doxycycline Treatment

*cp27.5* wild-type and mutant larvae were treated at 24 hpf with 50 µg/mL doxycycline (ClonTech Mountain View, CA, USA). Larvae were observed at 3 dpf to look for GFP expression in the CNS and imaged using a Nikon SMZ1500 epifluorescence stereoscope and Nikon NIS-Elements Version 3.1 software.

### 2.11. Fluorescent Tracer Injections

Wild-type and mutant *cp27.5* larvae were anesthetized in 0.02% tricaine and coinjected intravenously into the common cardinal vein with approximately 1.8 nL of 10 mg/mL 3-kDa Cascade blue dextran, 10-kDa rhodamine dextran, and a 40-kDa anionic fluorescein dextran (Invitrogen, Carlsbad, CA, USA). Injections were performed using a micromanipulator and a pneumatic picopump (World Precision Instruments, Sarasota County, FL, USA). Immediately after injection, larvae were embedded in 1.2% low-melting-point agarose (Invitrogen, Carlsbad, CA, USA) made in egg water. Larvae were imaged within 30 min postinjection on a Nikon C1Si confocal microscope and analyzed using Nikon EZC1 Version 3.91 software.

### 2.12. RNA-Seq Analysis

RNA was extracted as described before previously (see Section 2.4) from wild-type and mutant *cp27.5* larvae at 72 hpf. Samples were submitted to the Genome Sequencing Facility Hartwell Center for Bioinformatics and Biotechnology at St. Jude Children’s Research Hospital for Quality Control (QC) and quantification analysis and sequencing. Approximately 500 ng total RNA was used to generate libraries with the TruSeq RNA v2 kit according to the manufacturer’s instructions (Illumina, San Diego, CA). PolyA-selected RNAs from wild-type and mutant zebrafish were sequenced using Illumina HiSeq 2500. The raw 100-bp pair end reads were mapped to zebrafish genome (Zv9) using Spliced Transcripts Alignment to a Reference (STAR) [12]. Duplicated reads were marked using Picard (http://picard.sourceforge.net/command-line-overview.shtml). The detailed mapping statistics are listed in Table 1.

Considering genome duplication, FPKM (fragments per kilobase of exon per million fragments mapped) for each RefSeq gene was calculated as follows: first fragments mapped to each exon region of a gene, regardless of their genomic location, were marked and summed up as the gene counts. The final FPKM value for the gene was calculated as:FPKM = F_gene_ × 1000 × 1,000,000/(F_allexon_ × L_gene_)
where F_gene_ is the total fragments mapped to a gene, F_allexon_ is total fragments mapped to exonic region in Zv9, and L_gene_ is size of unique exonic bases for the gene. Differential expression analysis was done using purely fold change between the mutant and wild-type cells. Gene Ontology (GO) analysis was done using the Database for Annotation, Visualization and Integrated Discovery (DAVID) [13], with differentially expressed genes that have moderate expression level (FPKM > 0.1).

### 2.13. Quantitative Real Time PCR (qRT-PCR)

RNA was extracted from 9 *cp27.5* wild-type and 9 *cp27.5* mutant larvae and cDNA was synthesized using the same method as described in previously (see Section 2.4). RNA concentrations were normalized between wild-type and mutant. Samples were treated with DNAse I (Ambion Life Technologies, Austin, TX, USA) followed by treatment with 25 mM EDTA (Ambion). 20 ng of cDNA was used per reaction for quantitative real-time PCR (qRT-PCR). qRT-PCR was performed using SYBR^®^ Green PCR Master Mix (Life Technologies, Carlsbad, CA, USA) on a 7500 Fast Real Time PCR machine (Applied Biosystems, Foster City, CA, USA) and analyzed using 7500 Software v2.0.6. Relative gene expression was calculated using the Delta Delta CT (ddCt) method relative to the reference gene *β-actin*. Error bars were based on the mean of 3 rounds of qRT-PCR using standard error. The primer templates used for each gene are listed in Table 2.

### 2.14. Identification of Novel Junctions in cp27.5 Mutants

From the RNA-Seq data, *tp53* and *mdm2* were analyzed to identify novel junctions or alternative splice sites in *cp27.5* mutants. Junctions were identified using a CompBio pipeline. From there, the low count junctions were filtered and normalized using Limma-Voom. The criteria for selecting differential exon junctions included the average intensity (log2 RPM) > 2 and a log2ratio > 2. The reads mapped to exon junctions were extracted and normalized to counts per million (RPM). Different usage of each junction between the wild type and mutant sample was evaluated and candidate junctions were selected with fold change greater than 4 and average RPM of the 2 samples more than 4.

### 2.15. Data Availability

The strains available upon request included *Et*(*cp:EGFP*)*^sj2^* which express GFP in the CP. The *cp27.5* strain was no longer available. Appendix A are available at FigShare. Appendix A contains the list of genes with at least 1 read observed in either one of the samples (wild type or mutant), genes expressed where the FPKM > 0.1 in both samples, gene ontology for biological processes and molecular function, the Kyoto Encyclopedia of Genes and Genomes (KEGG), and a list of upregulated spliceosome-related genes.. The data is deposited at Figshare: https://doi.org/10.6084/m9.figshare.12980063.v1. Code and software used to generate the whole exome sequencing and RNA-Seq data can be found at:https://www.qiagenbioinformatics.com/product-downloads/;https://academic.oup.com/bioinformatics/article/29/1/15/272537;http://picard.sourceforge.net/command-line-overview.shtml;https://david.ncifcrf.gov/https://gsajournals.figshare.com/s/1fdefac9d4dd482f5e4f.

Appendix A contains the analysis of novel junctions for *tp53* and *mdm2* which is deposited at Figshare: https://doi.org/10.6084/m9.figshare.12980069.v1.

## 3. Results

### 3.1. Exome Sequencing and mRNA Rescue Identifies and Confirms Mutation in sart1

We previously performed a forward genetic screen to identify mutants in CP development. From this screen we identified a line, *cp27.5*, and mapped the mutation to Chromosome 21 (Figure 2A). To identify the gene and type of mutation, we performed WES on *cp27.5* mutants and their wild-type siblings. We identified a point mutation in *sart1* containing a G to A transition at the donor site of exon 12 (Figure 2B). Upon sequencing mutant cDNA, we identified the retention of intron 12 containing a TAA premature stop codon (Figure 2C,D).

To verify *sart1* as the causative gene in *cp27.5* mutants, a rescue experiment was done by injecting *sart1* mRNA into single-celled embryos. We screened 462 uninjected and 375 *sart1* mRNA injected larvae at 4 dpf to look for rescue. While overexpression of *sart1* mRNA in wild-type embryos did not cause any overt phenotype, *sart1* mRNA was able to rescue defects observed in the mutants such as abnormal choroid plexus development, heart edema, curved body, and larvae survived past 4 dpf (Figure 3A). Some survived as late as 8 dpf. From this screen, we quantified the results and found that, based strictly on phenotype, the number of mutants for uninjected controls was 23.3%, while mutant larvae from a spawn injected with *sart1* mRNA was 1.5% (Table 3).

To confirm that *cp27.5* mutants were among those rescued, we collected uninjected *cp27.5* wild-type and mutant larvae, and *sart1* mRNA-injected larvae and sequenced individual DNA. We sequenced a total of 90 larvae with 20% of those having a mutant genotype. Not surprisingly, we identified mutants with wild-type phenotypes as represented in Figure 3A. Some mutants appear to have complete rescue as shown in the lower right-hand panel, while others had slight heart edema, smaller head size, and somewhat larger CP compared to wild-type (presented in two upper panels under Rescue in Figure 3A). Sequencing results are quantified in Figure 3B. Based on the sequencing data, we identified 10% of mutants we scored as having a wild-type phenotype. The other 10% we scored as abnormal or having only a partial rescue. However, even partially rescued mutants were far more phenotypically normal compared to uninjected mutants (Figure 3A).

### 3.2. sart1 mRNA Is Maternaly Derived and Expressed in the Zebrafish Brain

To determine when *sart1* is expressed, we performed RT-PCR using cDNA from 2 hpf to 120 hpf wild-type samples. We observed *sart1* expression for each timepoint and also established that *sart1* is maternally derived based on its expression at 2 hpf (Figure 4A). These RT-PCR results confirmed whole-mount in situ hybridization data performed by Thisse and Thisse (2004) which presented *sart1* expression throughout the entire embryo during early cleavage [14]. As we observed phenotypes primarily within the head and eye region in *cp27.5* mutants, we wanted to determine whether *sart1* is spatially or temporally regulated within these regions or ubiquitously expressed. We looked at three developmental timepoints to determine *sart1* expression: 48 hpf when we begin to observe mutant phenotypes, 72 hpf, and 96 hpf when mutants begin to deteriorate. Using whole-mount in situ hybridization, we observed *sart1* localized in the head at 48 hpf (Figure 4B). This expression was consistent throughout development, although it appeared to become less prominent in this region by 96 hpf (Figure 4B). We also wanted to determine whether there were differences in expression between *cp27.5* wild-type and mutant. While *sart1* was expressed in the head at 72 hpf for both wild-type and mutant, *sart1* was more highly expressed in mutants compared to wild-type (Figure 4C). We did not contribute increased expression to increased permeability of the probe in mutants since the positive control, *deltaC*, was not intensified (Appendix A). Interestingly, *deltaC* expression, which is normally localized in the brain and eyes, was disordered and reduced in *cp27.5* mutants.

### 3.3. sart1 Mutants Have Altered Protein Expression in the Brain and Eye Along with Increased Activated Caspase 3

Because *sart1* expression was localized to the brain, we wanted to determine if other proteins in this region might be affected by *sart1* upregulation. Using IHC, we looked at Zinc Finger Protein (Zpr1), a marker that labels the pineal gland and photoreceptors in the retina. *cp27.5* mutants had little to no Zpr1 expression in the photoreceptors (Figure 5 Row 1). Counterstaining with DAPI revealed that retinal lamination was also absent. However, Zpr1 expression was present in the pineal gland in mutants indicating that not all Zpr1 expression was affected by the *sart1*mutation (Figure 6A,B). Another protein, Synaptic Vesicle Protein (SV2), which labels synaptic vesicles in the brain and eye, was also affected. Mutants had no SV2 expression in the eye (Figure 5 Row 2). However, SV2 was present in the brain and appeared to be slightly increased compared to wild-type.

Since we did not detect expression of Zpr1 and SV2 in the eye, and because mutations in splicing factors have previously been shown to contribute to retinal degeneration [15,16], we hypothesized that the cells in the eye may be undergoing cell death. To test this, we used an activated Caspase-3 antibody. While no activated Caspase-3 was detected in *cp27.5* wild-type larvae, it was significantly expressed throughout the eye of *cp27.5* mutants (Figure 5 Row 3). Caspase-3 was also detected in the dorsal region of the brain near the brain ventricle (Figure 5 Row 3 inset and Figure 6C–F). To determine if vasculature was affected, we used an antibody for Glut1, a glucose transporter specifically expressed in the CNS vasculature, to visualize blood vessels in the brain and eye. While mutants do have blood vessel formation, they are not as numerous and extensive as those seen in wild-types (Figure 5 Row 4). We also examined the neuronal marker HuC and S100 Calcium Binding Protein (S100β) which detect astrocytes. HuC could be detected in mutant brains, but little if any was present in the eye (Figure 5 Row 5). As the original line from *cp27.5* was generated using the Tetracycline-controlled transcriptional activation (Tet)-On system containing a neuron-specific promoter (HuC) driving GFP expression, we confirmed these results by treating wild-type and mutant larvae with doxycycline to induce neuron-specific GFP expression. As seen with the HuC antibody, the brain and spinal cord contain HuC expression in wild-type and mutant, but there is little to no expression in the eye of mutants (Figure 6G–J). For S100 β, we detected expression in the brain and eye of wild-type larvae, but detected little if any expression in mutants (Figure 5 Row 6).

### 3.4. sart1 Mutants Have Increased Permeability in the Brain Ventricle

Although there appeared to be a reduced number of blood vessels in the brain, we wanted to examine whether reduced vasculature led to increased permeability throughout the larva. We injected fluorescent tracers of different molecular weights intravenously into wild-type and mutant at 2 dpf to look for leakage. Because the increased permeability may be size-selective, we coinjected a 3-kDa cascade blue, a 10-kDa rhodamine, and a 40-kDa fluorescein dextran. While the 3-kDa and 10-kDa tracers leaked into the brain ventricle in wild-type, it did not penetrate into the brain itself or surrounding areas. In mutants, these tracers entered the brain ventricle, and leaked into nearby tissues (Figure 7A–D). In wild-type, the 40-kDa tracer was retained within the vasculature (Figure 7E,F). Because this tracer remained within the blood vessels, we were able to observe reduced vasculature throughout the brain and abnormal vascular patterning around the eye in *cp27.5* mutants. However, this phenotype was restricted only to these regions. Vascular patterning and permeability along the tail was normal (Figure 7G,H).

### 3.5. RNA-Seq Analysis Identifies Upregulated Apoptotic Genes

To determine what genes are affected by mutated *sart1*, we performed an RNA-Seq analysis to detect an up or downregulation of genes throughout the zebrafish genome by comparing *cp27.5* wild-type and mutant RNA expression levels. The complete analysis is found in Supporting Information in File S1. Because *sart1* is involved in spliceosome assembly, a process that occurs ubiquitously in all cell types, we expected to identify dysregulation of genes involved in a number of processes. Because the entire RNA-Seq data is too extensive to be described here, we have reported in the following paragraphs only the genes we confirmed to be up or downregulated by qRT-PCR (Table 4). The fold change normalized to β-actin for each gene is found in Figure 8A. Table 5 contains a more widespread look at the genes analyzed by RNA-Seq and their relative expression.

From the RNA-Seq analysis and qRT-PCR, we identified upregulation of *sart1* in mutants, thus confirming the whole mount in situ hybridization data showing elevated *sart1* expression in the brain compared to wild-type. As mentioned previously, Sart1 has been demonstrated to be involved in apoptosis where it appears to play a dual role by inducing cell cycle arrest [17], but also aiding in drug resistance in cancer cells [18]. Because of this, we also analyzed genes involved in cell cycle regulation. One of the more highly upregulated genes was *tp53*. While the *tp53* transcript was upregulated, we were also interested to see whether p53 protein was increased. By performing western blot analysis, we observed that both wild-type and mutant had a faint band at the predicted protein size of 42 kDa. However, in *cp27.5* mutants, a larger and more highly expressed band was detected closer to 50 kDa, and another band, while not as prominent, was detected to be greater than 50 kDa. These bands are faintly present in wild-type samples (Figure 8B). We also observed an increase expression in *mdm2*, which acts as an inhibitor of *tp53* by ubiquitinating p53 and targeting it for degradation. Further analysis of *tp53* and *mdm2* revealed novel junctions for both genes indicating that mutated *sart1* does affect splicing of apoptotic genes (Figure 9 and Appendix A).

In addition, we detected increased expression of *clusterin* (*clu*). Studies in zebrafish have observed *clu* expression specifically within the CP [19]. CLU has be associated with an increased risk of Alzheimer’s disease and plays a role in the clearance of amyloid (Aβ) peptide [20,21].

### 3.6. RNA-Seq Analysis Identifies Upregulated Spliceosome Components

Previous studies have identified increased expression of spliceosome transcripts when another component of the spliceosome itself, or a protein involved in spliceosome assembly is defective [10,15]. Because Sart1 is involved in the recruitment of U4/U6.U5 tri-snRNP to the spliceosome, we wanted to observe whether additional factors involved in spliceosome assembly were also defective or upregulated in order to compensate for possible defects in Sart1 function. The first gene confirmed by qRT-PCR was *prpf31*. This gene is a member of the precursor RNA processing (PRP) genes and, similar to *sart1*, plays a role in recruiting U4/U6.U5 tri-snRNP to the spliceosome [22]. Interestingly, mutations of *prpf31* are found in patients with RP, a degenerative disease of photoreceptors in the eye [22]. We determined that *prpf31* was upregulated in *cp27.5* mutants. We also observed upregulation of *lsm7*, a gene from another class of spliceosome components. Lsm7, which binds to U6 in the spliceosome, is a member of a family of proteins that are also involved in assembling major spliceosome factors [23].

### 3.7. RNA-Seq Analysis Identifies Upregulated Claudin 5a and Matrix Metalloproteinase Protein (MMP9)

In characterizing *cp27.5* mutants, we also observed defects in Claudin 5 (Cldn5) protein expression. Whole mount IHC showed Cldn5 expression on the surface of the brain ventricle in mutants, however, we did not observe co-localization with the CP as seen in wild-type larvae (Figure 8C). The results also suggested increased Cldn5 protein in mutants. Based on RNA-Seq and qRT-PCR data, we observed increased expression of *cldn5a* transcript in mutants, indicating that protein expression in the brain ventricle may also be upregulated. In a recent study, the enzyme MMP9 was suggested to lead to Cldn5 degradation at the blood-cerebrospinal fluid barrier (BCSFB) allowing for leukocytes entry into the cerebrospinal fluid [24]. MMP9 is also said to be upregulated in the CP in the presence of inflammation [25]. We observed a significant increase in *mmp9* expression in *cp27.5* mutants; however, whether upregulation is specific to the BCSFB requires further investigation.

### 3.8. RNA-Seq Analysis Identifies Downregulation of Vision-Related Genes

The final class of genes confirmed by qRT-PCR were those specific to the eye. Because retinal degeneration has been associated with splicing defects or mutations in spliceosome components [3,15], we were interested to see if mutations in *sart1* led to decreased expression of vision-related genes. Cone-rod homeobox (CRX) is a transcription factor found in photoreceptor cells and mutations have been associated with cone-rod dystrophy [26]. We observed a dramatic decrease in *crx* expression, which is not surprising due to our IHC data showing a loss of photoreceptors in *cp27.5* mutants. We also looked at *phosphodiesterase 6H* (*pde6h*), a gene that encodes for the γ (gamma) subunit of a cyclic guanosine monophosphate phosphodiesterase specific to cone photoreceptors. Mutations in this gene have been associated with achromatopsia, which is an inherited retinal dystrophy [27]. *pde6h* was also downregulated in *cp27.5* mutants. The final gene analyzed was *opsin-1*, *short-wave-sensitive 1* (*opn1sw1*), a gene specific to zebrafish expressed in ultraviolet cone photoreceptors [28]. Again, due to its specific expression in the retina, *opn1sw1* was downregulated in *cp27.5* mutants. While we provide a brief overview of specific genes identified from the RNA-Seq analysis and their response to the *sart1* mutation, future studies will examine a more in depth the role of Sart1 in regulating the expression levels and patterning of these genes. Suggested roles for Sart1 are described below.

## 4. Discussion

The current study describes a zebrafish mutant with a point mutation in *sart1* that results in abnormal CP development [6] in addition to defects throughout the brain and eye. In addition to splicing, *S*art1 has been implicated in a number of different processes such as cell cycle arrest and apoptosis and has been suggested as a target for gene therapy due to its role as an antigen recognized by cytotoxic T-lymphocytes (CTLs) in certain types of cancer [17]. More recently, *S*art1 has been shown to contribute to drug resistance in cancer cells [18,29]. While the goal of this study was to characterize *sart1* and identify other genes affected by this mutation, future experiments plan to discover the role of Sart1 in overall cellular processes such as cell death; in addition to better understanding its traditional role in splicing.

In our study, we identified the point mutation in *sart1* using WES. We were able to validate along with previously published studies [7,30] that WES is a suitable method for identifying genes in zebrafish. WES allowed us to effectively identify what gene contained the mutation and that the mutation was a transition. In future genetic mapping experiments, we plan to use WES as our main method for identifying the map position and type of mutation. Studies have cited that more than 9000 mutants have been discovered as a result of forward genetic screens in zebrafish, but many of the mutants have not been further studied and 62% of them are not cloned. This is due to limitations involved in positional cloning including the time, money, and effort to identify the gene [31]. Therefore, we propose that WES provides a more efficient method than positional cloning to identify mutants generated from forward genetic screens.

Additional validation to show that the phenotypes were due to a mutation in *sart1* was done by performing mRNA rescue experiments. *sart1* mRNA prevented forebrain necrosis at 48 hpf, which is the earliest detectable phenotype in *cp27.5* mutants. We were able to rescue phenotypes that began at 3–4 dpf such as heart edema, small head and eyes, reduced heartbeat, and circulation. From this analysis, a number of wild-type larvae were scored as abnormal. This could be due to a dominant negative effect of the *sart1* mutation. However, in Figure 2B representing the whole exome sequencing data, we have shown that the zebrafish larvae screened as “wild-type” are heterozygous because some of the fish in the sample had the G → A transition. We believe that if a dominant-negative mutation was present, the heterozygous fish would have similar phenotypes if not the same phenotypes, as the homozygous mutants because the phenotypes can be observed quite easily and at early developmental time points. We suggest that abnormal phenotypes for wild-type larvae may be due to off target effects from the injection process. When sequencing wild-type and mutant DNA from *sart1* mRNA injections, we identified only 20% as mutant rather than 25% according to Mendelian genetics. We attributed the decreased percentage due to damaged embryos from microinjection that may not have survived past early development. For uninjected samples, we observed between 23 and 25% mutant.

Prior to 48 hpf, mutants were undistinguishable from wild-type. The maternally derived transcript as identified by RT-PCR was present shortly after fertilization which may explain why we were unable to isolate mutant embryos prior to 48 hpf because of maternal contribution. Previous studies performed by Thisse and Thisse have also identified *sart1* expression shortly after fertilization during early cleavage thereby also confirming that the transcripts are maternal [14]. Interestingly, by whole mount in situ hybridization and RNA-Seq analysis, we determined that the point mutation in *sart1* results in upregulation of the transcript. Upregulation of mutated splicing genes have been demonstrated in other studies such as *prpf4*, which interacts with the U4/U6 di-snRNP and stabilizes the complex [15]. A mutation from a proline to leucine in *prpf4* led to its increased expression and also upregulation of other splicing factors including *sart1*. Similar to our study, they detected a loss of photoreceptors in the zebrafish eye and a downregulation of vision related genes such as *opn1lw1*, another opsin gene [15].

Additionally, the loss of expression in other vision-related proteins, such as Zpr1, SV2, HuC, and S100β, is not surprising due to previous studies showing that mutations in splicing factors can result in photoreceptor degeneration in diseases such as RP [15] where the most noticeable phenotype is a loss of photoreceptor cells [32]. Interestingly, for proteins expressed throughout the CNS, such as Zpr1 and SV2, there were differences in expression between the brain and eye suggesting that *sart1* may be regulating expression of certain genes in a tissue-specific manner. A recent study cited six genes related to splicing that have mutations in patients with RP including *PRPF31*, *PRPF8*, *PRPF3*, *PAP-1*, *SNRNP200*, and *PRPF6*. These mutations are responsible for autosomal dominant RP in more than 12% of cases [33]. Of these genes, three including *PRPF8*, *PRPF31*, and *PRPF3* are involved in forming and recruiting the U4/U6.U5 tri-snRNP to the spliceosome [32]. The exact cause of how mutations in splicing genes result in RP is still unknown, however, several models or hypothesis have been suggested by Mordes et al. [32]. Jia et al. also proposed a hypothesis where mis-splicing due to mutations in U2, a major spliceosome component, results in a buildup of mis-spliced mRNA that translates into dysfunctional proteins. Another hypothesis by Jia et al. proposes that introns retained within the mRNA sequence cause the spliceosome to remain at the intron rather than be released, resulting in alternative splicing and a toxic feedback loop where mRNA continues to be mis-spliced and cell death occurs [10].

In addition to splicing factors, mutations in other genes such as *crx* encoding a transcription factor, and *pde* genes involved in phototransduction, both of which were downregulated in *cp27.5* mutants, also contribute to RP [32]. We have to consider that perhaps the downregulation of vision-related genes is simply due to a developmental delay in these mutants. However, we do not believe that developmental delay can be contributing to the downregulation of genes in *cp27.5* mutants because the expression of these genes is not downregulated in every tissue. For example, mutants lacked Zpr1 expression in the retina, although expression was present in the pineal gland (Figure 6A,B) and SV2 exhibited similar expression in the brain between wild-type and mutants, but was decreased in the eye in *cp27.5* mutants. We would anticipate a decreased expression of these markers in all tissues had the downregulation been due to developmental delay.

In *cp27.5* mutants, we detected increased cell death within the eye as observed by activated Caspase 3 antibody staining. From the RNA-Seq data, we also detected upregulation of *tp53* and *mdm2*. This is interesting because Mdm2 acts as an inhibitor of p53, so we did not expect that both would be upregulated. While upregulation of *tp53* explains the increased cell death found in the eye and brain, increased expression of *mdm2* may indicate that it is trying to compensate for increased *tp53* expression. However, in previous studies, a feedback loop has been shown to occur where p53 can activate Mdm2 [34]. During cellular stress, the cell death pathway is initiated causing p53 levels to increase and Mdm2 levels to decrease. As a result of decreased Mdm2, p53 initiates *mdm2* transcription. As Mdm2 levels then increase, Mdm2 in turn acts to inhibit p53 [34]. Future studies will investigate whether *mdm2* is functional in inhibiting the p53 protein and whether p53 activates Mdm2 and forms a negative feedback loop. We also believe that the mutated *sart1* is affecting splicing of these apoptotic genes. We have identified novel splice site junctions in *p53* and *mdm2* in *cp27.5* mutants (Figure 9) suggesting that these novel splice junctions result in new protein isoforms that cannot be properly regulated. To confirm this hypothesis, a western blot analysis detected an alternative p53 protein synthesized in *cp27.5* mutants as identified by a prominent larger band for p53 than the predicted 43 kDa in mutant samples. While this band was also detected in wild-type samples, expression was much less compared to mutants. We believe this is due to heterozygous siblings present in the wild type sample. We suggest from our data that alternative splicing due to mutated Sart1 causes additional p53 proteins to be synthesized in *cp27.5* mutants that cannot be inhibited by Mdm2 thereby resulting in increased cell death (Figure 10). We also hypothesize that novel splice site junctions in *mdm2* may result in a defective protein that is unable to inhibit p53 function. As to why cell death is localized to the eye and brain has yet to be determined, but it is possible that Sart1 is affecting splicing in a tissue specific manner.

*cp27.5* was originally identified from a genetic screen isolating mutants with CP developmental defects. Interestingly, one of the genes we identified from the RNA-Seq data was *clu*. This gene is expressed specifically within the zebrafish CP in early development [19]. It acts as a chaperone and is a prosurvival gene in its secreted form by binding to Bax and preventing Bax from traveling to the mitochondria [35]. In its nuclear form, which originates from alternative splicing, it acts as a prodeath gene by promoting cell death in a Caspase-3 *independent* manner [35,36]. This agrees with our findings showing *clu* and Caspase-3 upregulation in *cp27.5* mutants. Future studies will investigate whether the nuclear form of *clu* is responsible for cell death in *cp27.5* mutants and whether cell death is specific to the CP. Upregulation of *clu* from the RNA-Seq analysis may be due to alternative splicing of *clu* resulting in its overexpression and therefore increased cell death. Alternatively, as mentioned previously, the secreted form of *clu* can act as a prosurvival protein [35]. If the secreted protein is upregulated in mutants, it may act in competition with proapoptotic genes such as *tp53* and compensating for cell death in the CP. However, even though *clu* was found to be upregulated at the transcript level, future studies will need to determine whether it is translated into a functional protein.

Another protein demonstrated in previous studies to be expressed in the CP is Cldn5 [6]. From the RNA-Seq data, we determined that *cldn5a* is also slightly upregulated in *cp27.5* mutants. While we were able to detect Cldn5 expression in mutants by whole mount IHC, future experiments will need to determine if it is functional. Cldn5 appeared to be mislocalized in mutants due to an absence of co-expression with GFP. Tracer studies revealed abnormal vasculature in the CNS indicating that the *sart1* mutation may be strictly CNS specific and does not affect peripheral tight junctions or vasculature.

While the genetic pathways downstream of *S*art1 in *cp27.*5 mutants have yet to be determined, the current study has revealed dysregulation of genes involved in apoptosis, splicing, vision, tight junctions, and matrix metalloproteinases. RNA-Seq analysis has provided substantial insights into genes and potential pathways regulated by Sart1. This study and others have demonstrated that mutations in splicing machinery or proteins associated with the spliceosome have widespread affects in other signaling processes in addition to splicing defects. Additionally, differences between phenotypes observed in the CNS versus the periphery may suggest that *S*art1 regulates the expression of certain genes in a tissue-specific manner.

## Figures and Tables

**Figure 1 cells-09-02340-f001:**
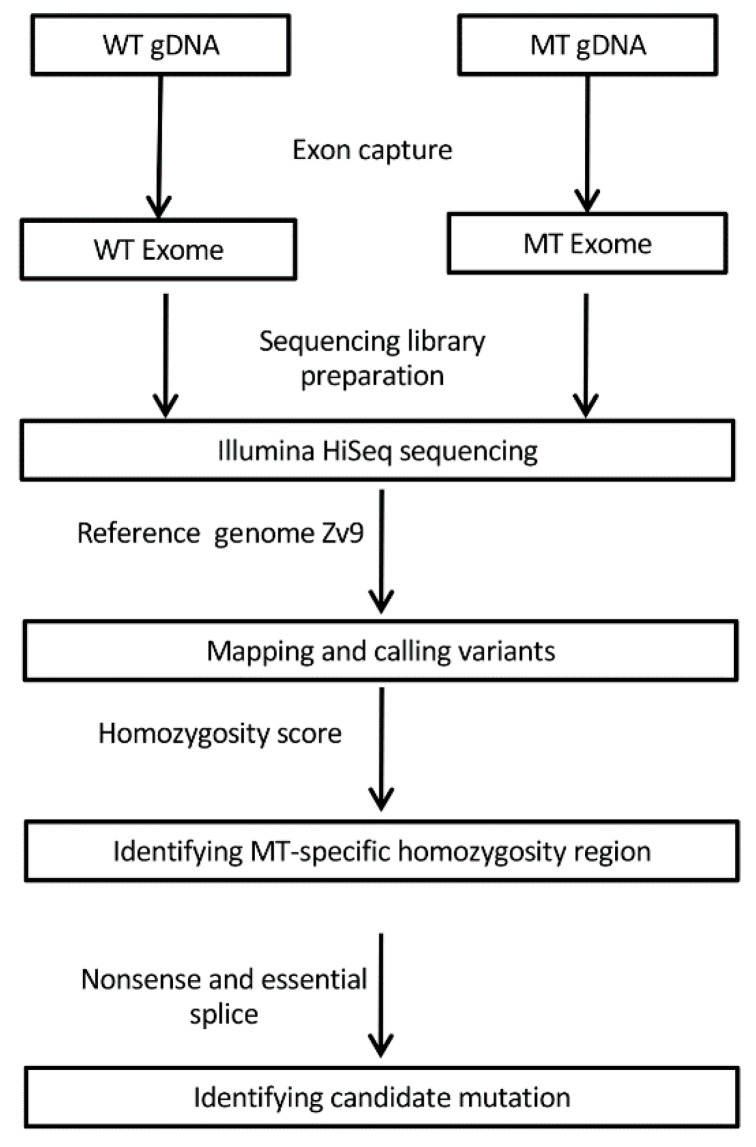
Schematic of whole-exome sequencing (WES) workflow and analysis. Exons were captured from wildtype (WT) and *cp27.5* mutant (MT) genomic DNA (gDNA) and analyzed to identify the candidate mutation.

**Figure 2 cells-09-02340-f002:**
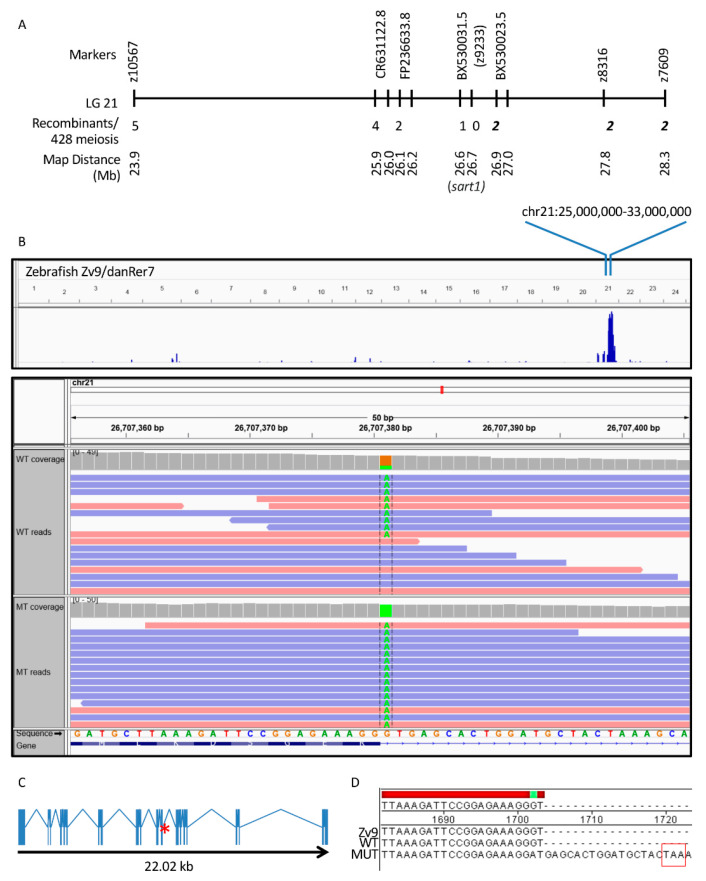
Positional cloning defines critical interval on chromosome 21 and WES identifies a point mutation in *sart1*. (**A**) A critical interval was defined on Chromosome 21 using positional cloning. For bulked segregant analysis, the polymorphic marker z9233 demonstrated linkage having zero recombinants identified out of 450 meiotic events. Additional markers were analyzed both proximal and distal to z9233 to define a critical interval of approximately 0.4 Mb. The numbers of recombinants on the proximal side are shown in regular type and those distal to z9233 are shown in bold italic font. *sart1* lies within the contig sequence BX530031.5. In (**B**), the blue peak is representative of the highest homozygosity score where mutants are homozygous within this defined loci along Chromosome 21. This finding is further demonstrated in the panel below where the reference genome contains a guanine (G) (in orange) at this position and all mutants have an adenine (**A**) (in green). An adenine is present in a portion of the wild-type sequence due to the fact that wild-type siblings may be heterozygous for the mutation. (**C**) shows that *sart1* contains a point mutation at the end of exon 12 (red asterisk). (**D**) DNA Sanger Sequencing of *cp27.5* wild-type and mutant cDNA confirms the G → A transition and reveals the retention of intron 12 and introduction of a premature stop codon (TAA).

**Figure 3 cells-09-02340-f003:**
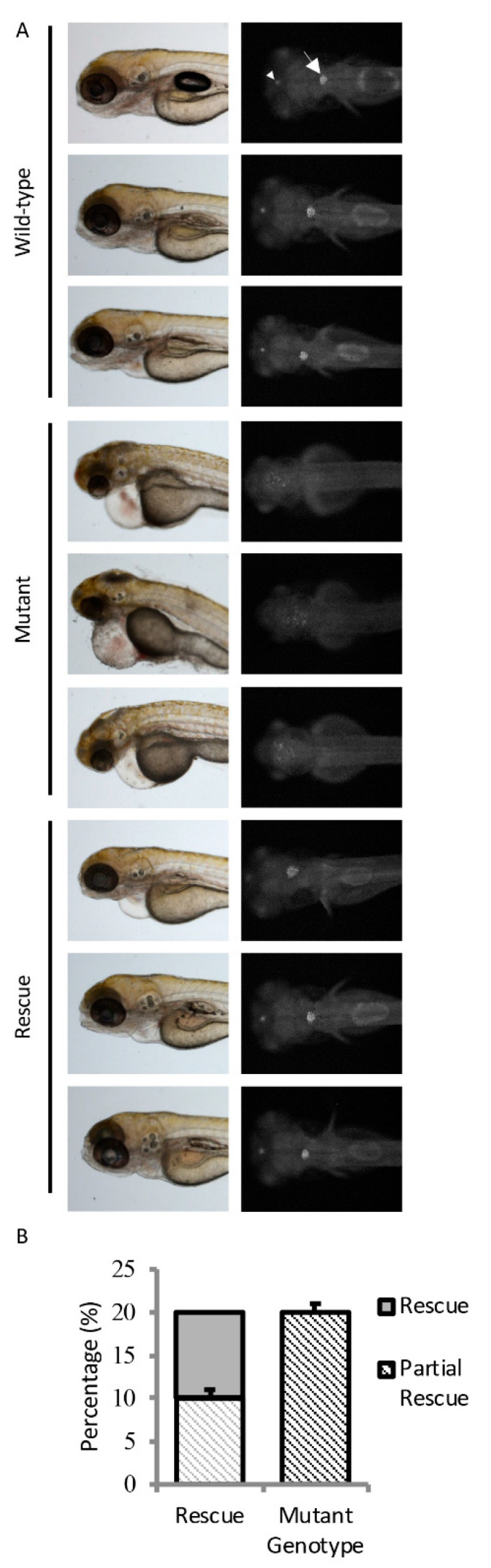
*sart1* rescues mutant phenotypes. (**A**) Images show three uninjected wild-type, three uninjected mutant, and three *sart1* injected mutants in the top, middle, and bottom panels, respectively. The left panel for all conditions is a bright field image showing the lateral view of zebrafish larvae at 4 dpf. The right panel for all conditions is a fluorescent image showing the dorsal view of zebrafish larvae at 4 dpf. The arrowhead represents the diencephalic choroid plexus (dCP) while the arrow represents the myelencephalic choroid plexus (mCP). Injection of *sart1* mRNA resulted in mutants with phenotypes similar to wild-type larvae. The three rescued mutants represent a range of phenotypes observed after *sart1* mRNA injection. Scale bars are 100 µm for bright field images at 3× magnifications and 50 µm for fluorescent images acquired at 9× magnifications. (**B**) Sequencing a total of 90 *cp27.5* wild-type and mutant larvae identified 20% with a mutant genotype. Error bars are based on the mean of 3 rounds of injections using standard error.

**Figure 4 cells-09-02340-f004:**
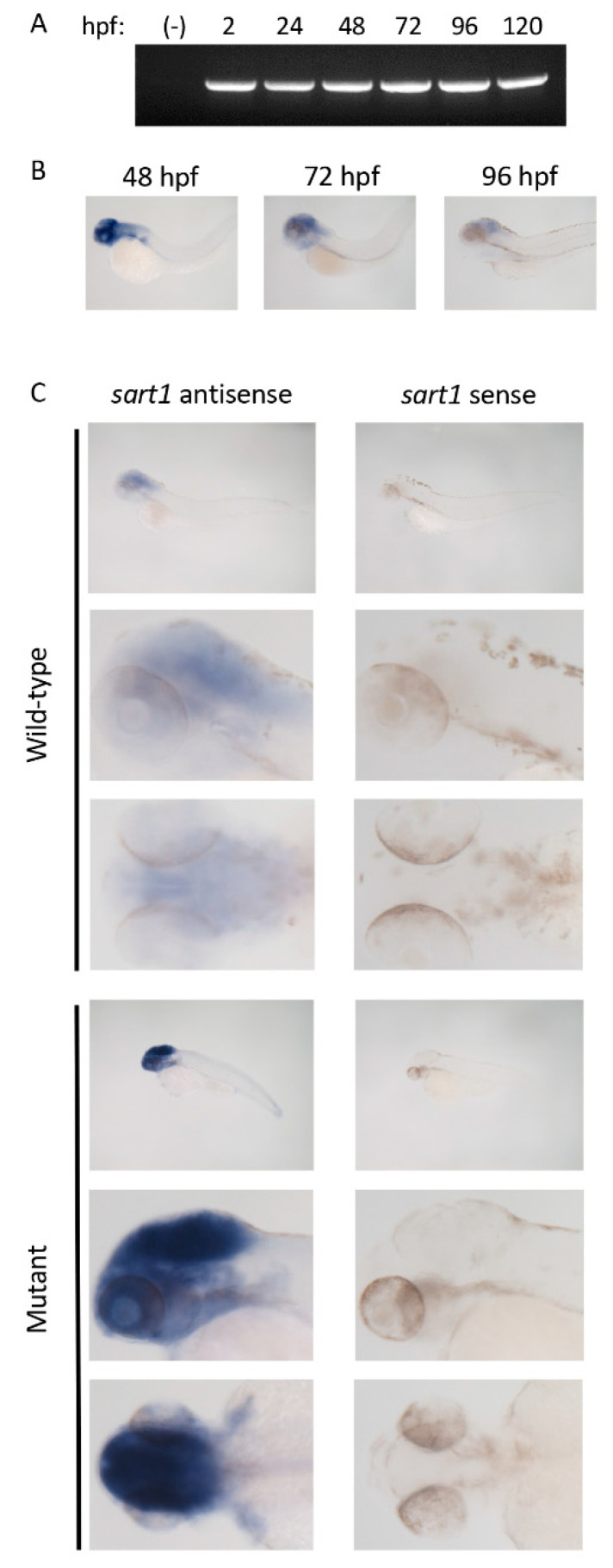
RT PCR and whole mount in situ hybridization identifies *sart1* expression. (**A**) RT-PCR shows *sart1* expression from 2 hpf to 120 hpf in zebrafish larvae including a water negative control (-). (**B**) Whole mount in situ hybridization shows *sart1* expression localized to the brain at 48 hpf, 72 hpf, and 96 hpf. (**C**) Whole mount in situ hybridization shows localization and relative expression levels of *sart1* in *cp27.5* wild-type and mutant larvae at 72 hpf. Images are ordered as lateral view at low magnification, lateral view at high magnification, and dorsal view at high magnification for both antisense (left column) and sense (right column). Scale bars are 50 µm.

**Figure 5 cells-09-02340-f005:**
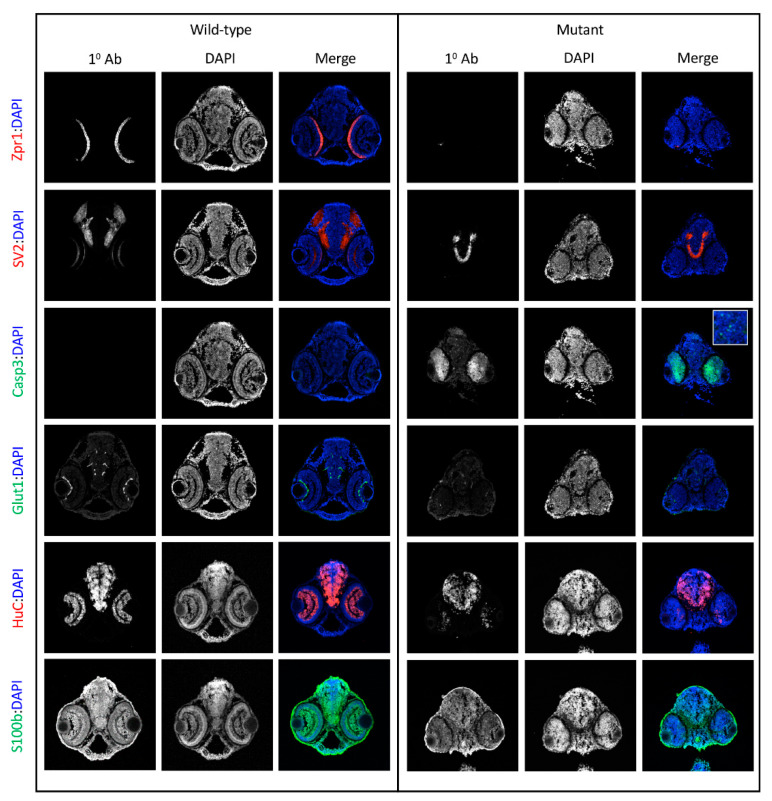
IHC characterizes *cp27.5* mutants. IHC identifies additional abnormalities in protein expression in *cp27.5* mutants at 3 dpf. All sections are counterstained with DAPI (blue) (middle columns for wild-type and mutant). **Row 1**: Zpr1 (red) labels photoreceptors in the retina which are present in wildtype, but not in mutant. **Row 2**: SV2 labels synaptic vesicles within the brain and within the inner plexiform layer of the retina in wild-type. SV2 is present in the brain, but absent in the retina in mutants. **Row 3**: Caspase 3 is not expressed in wild-type, but is prominent in the eyes in mutants. Mutant brains also contain punctate Caspase 3 expression (inset). **Row 4**: Glut1 is noticeably expressed in brain and eye vasculature in wild-type, but mutants have faint Glut1 expression in brain vasculature and eye. **Row 5**: HuC is expressed in neurons throughout the brain and eye in wild-type. HuC is expressed in the brain in mutants; however, it is not as organized as wild-type and only remnants of HuC are detected in mutant eye. **Row 6**: S100β labels astrocytes in the brain and eyes in wild-type, but is greatly reduced and sporadic in the brain and eyes of mutants. Scale bars are 50 µm.

**Figure 6 cells-09-02340-f006:**
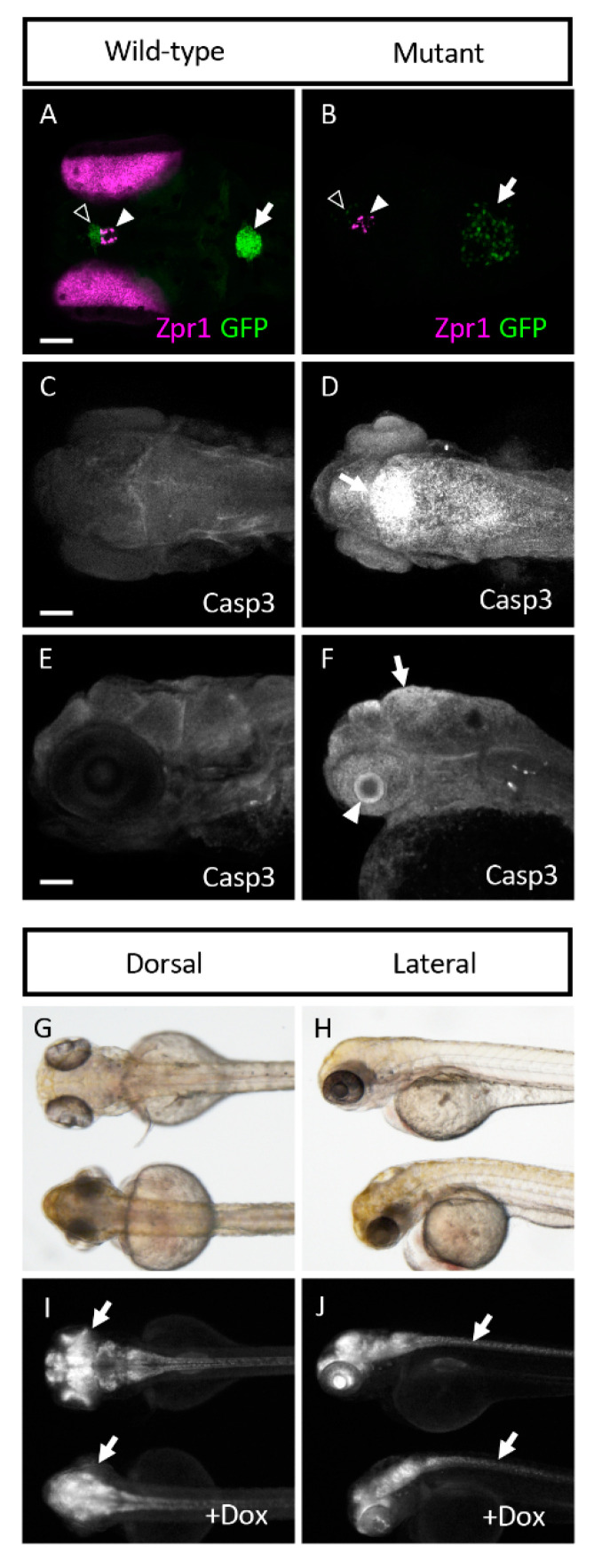
*cp27.5* mutants express Zpr1 in the pineal gland, activated Caspase 3 in brain ventricle and eyes, and HuC in brain and spinal cord. (**A**) Wild-type *cp27.5* larvae at 4 dpf with GFP (green) in the dCP (open arrowhead), and mCP (arrow) and Zpr1 expression (purple) in the pineal gland (filled arrowhead) and eyes. (**B**) Mutant *cp27.5* larvae at 4 dpf with GFP (green) in the dCP (open arrowhead), and mCP (arrow) and Zpr1 expression (purple) in the pineal gland (filled arrowhead). Mutants have no Zpr1 expression in the eye. dCP, diencephalic choroid plexus; mCP, myelencephalic choroid plexus. (**C**,**D**) Dorsal confocal image with activated Caspase 3 antibody reveals expression in the dorsal midline in *cp27.5* mutants, but not in *cp27.5* wild-type. (**E**,**F**) Lateral confocal image with activated Caspase 3 antibody reveals expression on the brain ventricle surface (arrow) and eye (arrowhead) in *cp27.5* mutants, but not in *cp27.5* wild-type. *cp27.5* larvae treated with doxycycline induces the TetON system using the HuC promoter resulting in GFP expression throughout the CNS. (**G**,**H**) Brightfield DIC images show wild-type larvae (top) and mutant larvae (bottom). (**I**) Wild-type larvae (top) treated with doxycycline express GFP eyes, while mutant larvae have no GFP expression in the eyes (arrows). (**J**) Wild-type larvae (top) and mutant larvae (bottom) treated with doxycycline both have GFP expression in the spinal cord (arrows). Scale bars are 50 µm.

**Figure 7 cells-09-02340-f007:**
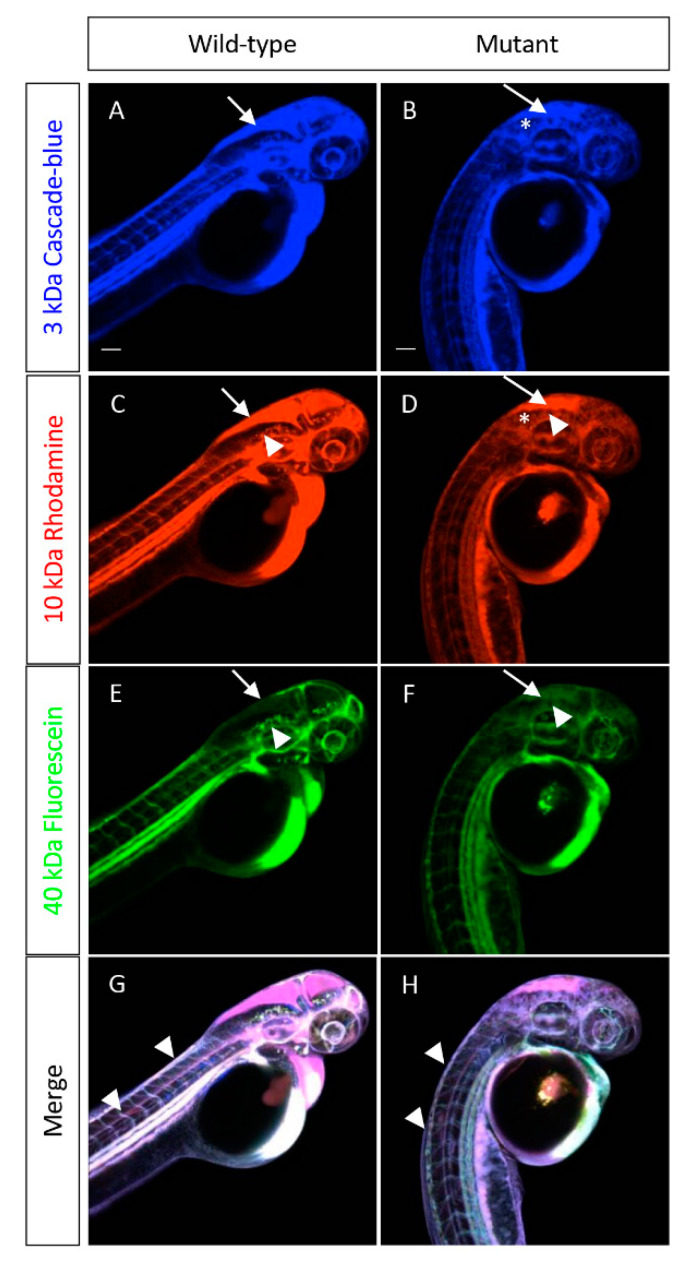
*cp27.5* mutants express Zpr1 in the pineal gland, activated Caspase 3 in brain ventricle and eyes, and HuC in brain and spinal cord. *cp27.5* wild-type and mutant larvae were intravenously injected with a mixture of a 3-kDa Cascade blue dextran, 10-kDa Rhodamine dextran, and 40-kDa Fluorescein dextran at 2 dpf. (**A**) The 3-kDa Cascade blue dextran is permeable to the brain ventricle in wild-type (arrow), but is restricted from the brain and eye (asterisks). (**B**) The 3-kDa Cascade blue dextran is retained in the brain ventricle (arrow), but also leaks into the surrounding tissue (asterisks) in mutants. (**C**) The 10-kDa Rhodamine dextran is permeable to the brain ventricle (arrow), but is also restricted from the brain and eye (asterisks) in wild-type. However, the 10-kDa Rhodamine dextran is more restricted to the brain vasculature compared to the 3-kDa Cascade blue (arrowhead). (**D**) The 10-kDa Rhodamine dextran is permeable to the brain ventricle (arrow), but also leaks into the surrounding tissue (asterisks) and is not restricted to the brain vasculature in mutants (arrowhead). (**E**) The 40-kDa Fluorescein dextran has little to no tracer leakage into the brain ventricle (arrow) and is restricted from entering the brain and eye (asterisk) in wild-type. The tracer is also retained within the brain vasculature (arrowhead). (**F**) In mutants, the 40-kDa Fluorescein dextran is slightly permeable to the brain ventricle (arrow), but not as severe compared to the 3-kDa Cascade blue and 10-kDa Rhodamine dextran. The tracer does not appear to leak into the brain, eye, or surrounding tissue as severely as the smaller tracers; however, there is still absence of the tracer retained within the brain vasculature (arrowhead). (**G**) Merged image of wild-type larvae coinjected with all three tracers. (**H**) Merged image of mutant larvae coinjected with all three tracers. Tracers are retained within the peripheral tail vasculature in mutants comparable to wild-type (arrowheads). Scale bars are 50 µm.

**Figure 8 cells-09-02340-f008:**
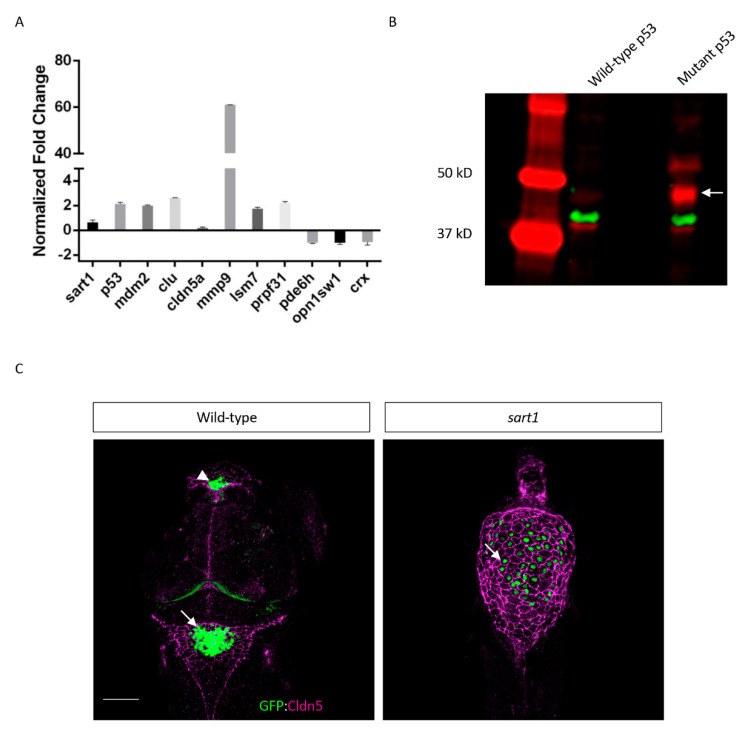
Multiple genes are upregulated or downregulated in *cp27.5* mutants along with abnormal expression of p53 and claudin 5 proteins. (**A**) RNA-Seq analysis was confirmed using qRT-PCR. Fold changes were normalized to β-actin which was set to zero. Positive values represent upregulated expression while negative values represent a downregulation of gene expression. The majority of genes are upregulated including *sart1*, apoptotic genes (*tp53*, *mdm2*, *clu*), tight junctions (*cldn5a*), extracellular matrix enzymes (*mmp9*), and spliceosome factors (*lsm7*, *prpf31*), while vision-related genes are downregulated (*pde6h*, *opn1sw1*, *crx*). Error bars are expressed as mean ± SE (standard error) for *n* = 3 from mutant samples. (**B**) Both *cp27.5* wild-type and mutant express p53 with a protein size of 42 kDa (red). However, *cp27.5* mutants also highly express a p53 protein that is absent in wild-type (arrow). Protein levels were normalized to β-actin (green). (**C**) Whole mount IHC was performed on *cp27.5* wild-type and mutant larvae at 3 dpf. In wild-type, the dCP (arrowhead) and mCP (arrow) express GFP (green). Claudin 5 (purple) is localized to the surface vasculature and co-localizes with the CP epithelia (yellow). In mutant, the dCP is absent and the mCP consists of individual sporadic GFP positive cells (arrow). Claudin 5 is expressed throughout the brain ventricle and surrounds CP epithelia, but does not colocalize with the cells as seen in wild-type. Scale bar is 50 µm.

**Figure 9 cells-09-02340-f009:**
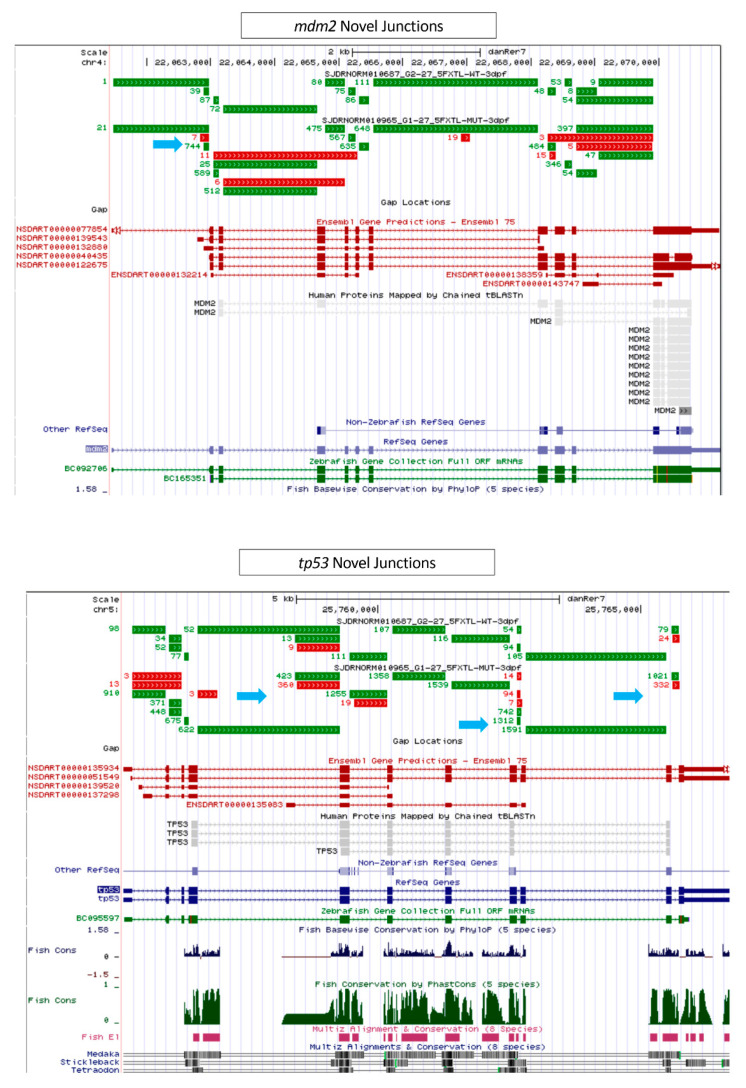
Novel junctions for *tp53* and *mdm2* in *cp27.5* mutants. Top panel: Junction read counts at *mdm2* locus displayed in the University of California Santa Cruz (UCSC) genome browser. Green represents known junctions and red represents novel junctions. Bottom panel: Junction read counts at *tp53* locus displayed in UCSC genome browser. Green represents known junctions and red represents novel junctions.

**Figure 10 cells-09-02340-f010:**
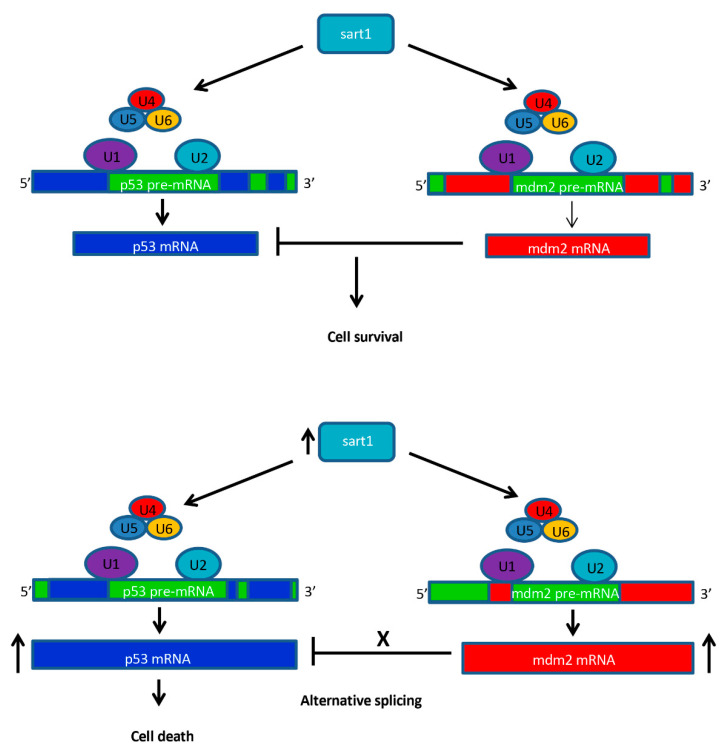
Schematic diagram for the hypothetical role of Sart1 in apoptosis. In wild-type, Sart1 acts to recruit the U4/U6.U5 tri-snRNP to the spliceosome complex where U1 and U2 are already bound to pre-mRNA at the 5′ splice site and branch point site, respectively. Once the spliceosome is assembled, it processes pre-mRNA by removing introns and producing mature mRNA transcripts for genes such as *tp53* and *mdm2*. Mdm2 inhibits p53 allowing for cell survival of photoreceptors. In mutants, while the *S*art1 transcript is upregulated, altered Sart1 protein may result in defective recruitment of U4/U6.U5 to the spliceosome thereby causing alternative splicing of *p53* and *mdm2*. Alternative splicing may result in altered proteins such as Mdm2 which can no longer function to inhibit p53 activity. In addition, alternatively spliced *tp53* transcripts may result in proteins that can no longer be inhibited by Mdm2. This phenomenon may lead to up-regulation of p53 and increased cell death in photoreceptors.

**Table 1 cells-09-02340-t001:** Mapping statistics of RNA-Seq data.

Sample	Reads	Mapped	NonDupMapped	Mpd% ^1^	Dup% ^2^
*cp27.5* Wild-type	61515460	58536062	45999884	95.16	21.42
*cp27.5* Mutant	99288072	94481824	74744636	95.16	20.89

^1^ percentage of reads mapped; ^2^ percentage of nonduplicate reads mapped.

**Table 2 cells-09-02340-t002:** Primer sequences for qRT-PCR.

Gene	Primer	Primer Sequence (5’–3’)	Tm	GC%	Product Length
*sart1*	Forward	GTCGCAAACTCGCCAAAGAG	60.1	55	127
	Reverse	TATGCTTGCCGCTTCTCCTG	60.5	55	
*p53*	Forward	CTCTCCCACCAACATCCACT	59.0	55	178
	Reverse	ACGTCCACCACCATTTGAAC	58.7	50	
*mdm2*	Forward	AACCGAGGCAGACTACTGGA	60.3	55	105
	Reverse	TCTGGAAGCCAATCAGCTCG	60.1	55	
*clu*	Forward	ACACCTCAAGTCTGCTCGAC	59.7	55	115
	Reverse	CCTTTGGACATCACTGCCTG	58.8	55	
*cldn5a*	Forward	ATGCTGTCTGGCTGACCAAA	59.9	50	190
	Reverse	CTTTCTGTTTTCGACGCGCT	59.8	50	
*mmp9*	Forward	AGACTTGGAGTCCTGGCGTT	61.1	55	147
	Reverse	ACGCTTCAGATACTCATCCGCTA	61.3	48	
*lsm7*	Forward	ACATGCGAGATCCTGATGACC	59.9	52	84
	Reverse	CAACAGACGTCCCTCGACAA	60.0	55	
*prpf31*	Forward	GTCAAGCAGGTCAAGCCTCT	60.0	55	221
	Reverse	GCTTGTCTGACTCTGCCACT	60.0	55	
*pde6h*	Forward	GACCACTCGCACCTTCAAGA	60.0	55	99
	Reverse	ACAGTGATGTCTGTGCCGAG	60.0	55	
*opn1sw1*	Forward	CGATTGCAGGTCTTGTGACG	59.6	55	195
	Reverse	GACCCTCGGGAATGTATCTGC	60.3	57	
*crx*	Forward	CCATTATGCTGTGAACGGGT	58.3	50	84
	Reverse	CTCGGAGTGGCTGGGTA	57.5	65	
*actb1*	Forward	TGAATCCCAAAGCCAACAGAG	58.5	48	150
	Reverse	TCACACCATCACCAGAGTCC	59.0	55	

**Table 3 cells-09-02340-t003:** *sart1* mRNA rescues mutant phenotypes.

Samples	Total Number Screened	Morphologically Mutant (%)
No Injection Control	462	23.3%
*sart1* mRNA	375	1.5%

**Table 4 cells-09-02340-t004:** Upregulated and downregulated genes confirmed by qRT-PCR.

Gene	Accession Number	log_2_(MUT-WT) from RNA-Seq	Fold Increase from qRT-PCR	Normalized for qRT-PCR
*sart1*	NM_001002673	3.35	1.64	0.64
*tp53*	NM_131327	4.22	3.14	2.14
*mdm2*	NM_131364	4.07	2.99	1.99
*clu*	NM_200802	3.76	3.61	2.62
*cldn5a*	NM_213274	3.42	1.20	0.20
*mmp9*	NM_213123	8.38	61.99	61.01
*prpf31*	NM_200504	3.71	2.75	1.76
*lsm7*	NM_001048006	3.71	3.22	2.23
*pde6h*	NM_200785	−16.92	0.00	−1.00
*opn1sw1*	NM_131319	−7.29	0.00	−1.00
*crx*	NM_152940	−1.71	0.05	−0.95
*actb1*	NM131031	0.71	1.00	0.00

**Table 5 cells-09-02340-t005:** Top 10 upregulated and downregulated genes from RNA-Seq with FPKM > 0.1 in both samples.

Name	Gene	Chromosome	log_2_(MUT-WT)
*fibronectin 1b*	*fn1b*	1	4.6840
*TIMP metallopeptidase inhibitor 2b*	*timp2b*	3	4.6815
*jun B proto-oncogene b*	*junbb*	3	4.5556
*nuclear factor of kappa light polypeptide gene enhancer in B-cells inhibitor, alpha a*	*nfkbiaa*	20	4.5268
*insulin-like growth factor binding protein 1a*	*igfbp1a*	20	4.4076
*CCAAT/enhancer binding protein (C/EBP), delta*	*cebpd*	24	4.2910
*tumor protein p53*	*tp53*	5	4.2233
*hepcidin antimicrobial peptide*	*hamp*	16	4.1544
*MDM2 oncogene, E3 ubiquitin protein ligase*	*mdm2*	4	4.0737
*clusterin*	*clu*	20	3.7648
*fibroblast growth factor binding protein 2b*	*fgfbp2b*	1	−2.3904
*parvalbumin 8*	*pvalb8*	3	−2.5884
*transcobalamin like*	*tcnl*	5	−2.7405
*guanine nucleotide binding protein (G protein), gamma transducing activity polypeptide 1*	*gngt1*	19	−2.7876
*actinodin4*	*and4*	14	−3.0036
*creatine kinase, mitochondrial 2a (sarcomeric)*	*ckmt2a*	10	−3.1316
	*zgc:73075*	19	−3.3992
	*zgc:13622*	7	−3.5883
	*zgc:73359*	3	−3.8776
*matrilin 1*	*matn1*	19	−5.4303

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
