# Peer review of "A *sart1* Zebrafish Mutant Results in Developmental Defects in the Central Nervous System"

_cells, 2020, doi:10.3390/cells9112340_

Round 1
Reviewer 1 Report
It is well designed study. I recommend to shorten a little bit text, its very long and might be easily cut (especially methods section). I have detected some inconsistencies in using abb and typos. Please replace also figures, they are vey low quality and some of them are not readable.
Reviewer 2 Report
The work presented by Henson and Taylor described the identification of Sart1 mutation to be responsible for the phenotype of the cp7.5 mutant. Furthermore, the authors provide a significant characterization of the mutants, both at the phenotypic level and transcriptome level. These analysis reveal vasculature leakage associated with the mutation. Furthermore, also preliminary, potential eye defects were also observed. The writing of the manuscript and the current method of data/figure presentation makes the manuscript difficult to read. The manuscript will benefit from significant edits. The following list highlight changes that must be addressed.
- Abbreviations should be described when first appear in text. Example: Line 40. dpf should be described as days post fertilization (dpf)
- Is a panel 3A showing images of a particular transgenic zebrafish? It is not clear what tissue is shown.
- Inconsistency in labeling of sart1. It is italicized in some places and not in others.
- relabeling of RT-PCR is suggested. It is not clear what day 0 is. How many hours post fertilization?
- Figure 5 can be supplemented. Its placement distracts from the main article.
- Labels in Figure 6 is not clear. There are no individual panels labeling but the text and legends refer to individual panels.
- GLUT1 staining in Figure 6 is not very convincing alone. Recommend costaining in a vasculature transgenic fish (FLK:EGFP or FLI:EGFP).
- Figures 6 and 7 is difficult to follow in text. Recommend rearrangement of figures to follow order in text.
- Figure 9B should have lanes labeled.
- Does injection of Sart1 mRNA rescue the staining or vasculature phenotype seen?
Reviewer 3 Report
In this paper, Henson and Taylor characterized a sart-1 mutant in zebrafish Danio rerio that was previously identified from an ENU mutagenesis screen identifying 80 mutants in choroid plexus development. By using different experimental approaches, genetic WSE sequencing, microinjection experiment to rescue mutants phenotype, immunocytochemistry, wholemount ISH, and RNA-seq the authors give a very nice complete description of this mutant. In particular, they identify several genes involved in apoptosis and splicing events that result upregulated in this mutant. Moreover, several defects involving abnormal choroid plexus development and the brain and eye are reported.
The paper is well presented, and the results are supported by the data shown.
I believe that this work presents a story that could be suitable for publication with minor revisions.
In particular, I suggest adding a Figure showing sart-1 mRNA expression by whole-mount ISH of a zebrafish egg to support the maternal expression. Moreover, in the RT-PCR experiments, I would like to see a normalizer gene (e.g. beta-actin) showing expression at all developmental stages.
Round 2
Reviewer 2 Report
The major concerns of this reviewer has been addressed.